# Recombinant GPEHT Fusion Protein Derived from HTLV-1 Proteins with Alum Adjuvant Induces a High Immune Response in Mice

**DOI:** 10.3390/vaccines11010115

**Published:** 2023-01-03

**Authors:** Hamid Reza Jahantigh, Angela Stufano, Farhad Koohpeyma, Vajihe Sadat Nikbin, Zahra Shahosseini, Piero Lovreglio

**Affiliations:** 1Interdisciplinary Department of Medicine, Section of Occupational Medicine, University of Bari, 70214 Bari, Italy; 2Department of Pathology, Faculty of Medicine, Emory University, Atlanta, GA 30033, USA; 3Shiraz Endocrinology and Metabolism Research Center, Shiraz University of Medical Sciences, Shiraz 7134814336, Iran; 4Department of Bacteriology, Pasteur Institute of Iran, Tehran 1316943551, Iran; 5Department of Medical Biotechnology, School of Allied Medical Sciences, Iran University of Medical Sciences, Tehran 1449614535, Iran

**Keywords:** HTLV-1, HAM/TSP, reverse vaccinology, multi-epitope vaccine, recombinant protein

## Abstract

The human T-cell leukemia virus type 1 (HTLV-1) is a positive single-stranded RNA virus that belongs to the delta retrovirus family. As a result, a vaccine candidate that can be recognized by B cells and T cells is a good candidate for generating a durable immune response. Further, the GPEHT protein is a multi-epitope protein designed based on the *Gag*, *Pol*, *Env*, *Hbz*, and *Tax* proteins of HTLV-1. In developing a suitable and effective vaccine against HTLV-1, the selection of a designed protein (GPEHT) with the formulation of an alum adjuvant was conducted. In this study, we assessed the potential of a multi-epitope vaccine candidate for stimulating the immune response against HTLV-1. In assessing the type of stimulated immune reaction, total IgG, IgG1, and IgG2a isotypes, as well as the cytokines associated with Th1 (IFN-γ), Th2 (IL-4), and Th17 (IL-17), were analyzed. The outcomes showed that the particular antisera (total IgG) were more elevated in mice that received the GPEHT protein with the alum adjuvant than those in the PBS+Alum control. A subcutaneous vaccination with our chimera protein promoted high levels of IgG1 and IgG2a isotypes. Additionally, IFN-γ, IL-4, and IL-17 levels were significantly increased after spleen cell stimulation in mice that received the GPEHT protein. The immunogenic analyses revealed that the GPEHT vaccine candidate could generate humoral and cell-mediated immune reactions. Ultimately, this study suggests that GPEHT proteins developed with an alum adjuvant can soon be considered as a prospective vaccine to more accurately evaluate their protective efficacy against HTLV-1.

## 1. Introduction

The human T-cell leukemia virus type 1 (HTLV-1) is a positive single-stranded RNA virus that belongs to the delta retrovirus family and contaminates around 20 million individuals globally [1,2]. The HTLV-1 infection is endemic in numerous countries in Africa, Central and South America, and some parts of the Middle East, and its main etiological agent is HTLV-1-Associated Myelopathy/Tropical Spastic Paraparesis (HAM/TSP), HTLV-1-associated Uveitis, and Adult T Cell Leukemia/Lymphoma (ATL) [3,4,5]. Further, HTLV-1 can be transmitted via blood transfusion, drug injection, sexual activity, and breastfeeding [6,7].

The genome of the HTLV-1 virus has common retroviral genes consisting of *gag*, *pol*, *pro*, *env*, and PX areas [8,9]. The *gag* gene encodes different proteins for releasing virus fragments [10]. Additionally, the *pro* and *pol* genes are essential for transcription and protease functions [11]. The pX area consists of four ORFs and encodes proteins such as *Tax* and bZIP factors (*HBZ*), which are essential for the pathogenesis of HTLV-1 and lowering HTLV-1 immunogenicity [12,13]. For instance, in 2012, Zhao et al. discovered that HBZ could inhibit IFN-γ responses and prompt the production of TGF-β and Foxp3 transactivation elements in the development of T regulatory cells as the reservoirs of HTLV-1 throughout infection, thus boosting the expansion of HTLV-1 infection by reducing CD4+ T cells and CTL activity [14,15]. Additionally, according to an in vivo study, inhibition of HBZ activity leads to a decrease in the proviral load and the absence of antibody response against HTLV-1 [16]. It was also revealed that HBZ mRNA is present in 100% of ATLL lymphocytes [17]. Additionally, Tax initiates ATLL pathogenesis along with HBZ [18]. The previous report further clarified that the Tax is very important for viral replication as it suppresses cell checkpoints and initiates the development of adult T-cell leukemia [19].

The pathogenesis of HTLV-1 starts with membrane fusion between the virus and CD4+ T cells [20]. The cellular connection is promoted by heparan sulfate proteoglycan [21]. Additionally, glucose transporter 1 (GLUT1) and neuropilin-1 were observed to play roles in cellular attachment [22,23].

As a result of the rising prevalence of HTLV-1, considerable research has been performed to explore efficient treatments for HTLV-1 infection [24]. Peptide-based vaccines were found to induce cellular and humoral immunity and are chemically safe without oncogenic capacity. For example, Frangione-Beebe et al. (2001) reported the peptide generated from glycoprotein 46 (gp46) of HTLV-1 could stimulate a high immune reaction against HTLV-1 in mice [25]. In addition, Sundaram et al. (2004) generated immunogenic peptides consisting of glycoprotein (gp21) subunits with T-cell epitopes from tetanus toxoid [26]. The generated peptides created high antibody responses in mice and prevented HTLV-1 infection. Additionally, Kazanji et al. (2006) generated a recombinant vaccine including gp46 and *Tax* fragments [27]. This peptide increased antibodies and IFN-γ responses, triggering high cellular immunity and a substantial decrease in the proviral load. Nonetheless, there is still no information regarding the future use of these vaccines among the general populace that is potentially exposed to this infection.

In our previous report [28], we used in silico strategies to forecast the epitopes that could potentially bind human T and B cells. In aiming to use a reverse vaccinology approach for the first time to identify immunogenic vaccine candidates against HTLV-1 while considering all possible epitopes of the five vital proteins, *Hbz*, *Tax*, *Pol*, *Gag*, and *Env*, a GPEHT recombinant vaccine was created against HTLV-1 that can be identified by T lymphocytes (MHCI and MHCII) and B cells to generate cellular and humoral responses in mice. Further, to develop a suitable and effective vaccine against HTLV-1, a protein vaccine candidate was selected with the formulation of an alum adjuvant. In this study, we expressed the GPEHT protein in the *E. coli* BL21 (DE3) strain and purified the protein using affinity chromatography. Subsequently, the immunogenicity of two doses of 20 and 40 micrograms (µg) of the GPEHT protein was evaluated in a mouse model. Finally, we analyzed the humoral and cellular responses elicited after vaccination.

## 2. Materials and Methods

### 2.1. Vaccine Engineering

A newly designed GPEHT hybrid protein was composed of the Gag (BAX35073.1), Pol (BAX35065.1), Env (AAA45389.1), Hbz (BAX35088.1), and Tax (AYN25375.1) genes from the genome of HTLV-1, according to relevant bioinformatics studies [28]. In this study, we joined the GPEHT protein domains together using GGGGS linkers. An evaluation of the characteristics of antigenicity and allergenicity was conducted to perform protein structure validation.

### 2.2. Plasmid Construction, Expression, and Purification of GPEHT Protein

The recombinant gene of GPEHT (~70 kDa) was built in a pET28a vector (cloning site of NcoI at the 5′ end and XhoI at the 3′ end) manufactured by Biomatik (Cambridge, ON, Canada). Furthermore, a hexahistidine tag (6xHis-tag) was included at the C-terminal of the GPEHT protein. The GPEHT vector was transformed into competent *E. coli* BL21 (DE3) cells via the heat shock technique. Expression of the recombinant proteins was caused by including lactose at final concentrations of 10 g/L in the LB broth medium. The protein expression was analyzed via 12% SDS-PAGE and validated using Western blot evaluation with an HRP-conjugated His-specific antibody (Sigma, St. Louis, MO, USA) at a dilution of 1:1000. Further, the concentration analysis of GPEHT protein in images from the SDS-PAGE and Western blot was performed using imageJ.

### 2.3. Protein Purification and Dialysis

The GPEHT protein was further purified under denatured conditions using a nickel-nitrilotriacetic acid (Ni-NTA) affinity chromatography column (Qiagen, Hilden, Gemrany), according to the Qiagen protocol (Qiagen and Valencia, 2003). Endotoxins were removed from the GPEHT protein using ε-poly-L-lysine-agarose (Pierce High-Capacity Endotoxin Removal Spin Column, 0.5 mL, #88274; Thermo Fisher Scientific, Inc., Waltham, MA, USA). The dialysis membrane was prepared via boiling in 1% acetic acid and EDTA (1 mM). Additionally, the boiling of the sample was repeated a second time in distilled water. The dialysis membrane was filled with purified protein and put in PBS solution for 24 h at 4 °C. The PBS solution was changed every 8 h. The samples were passed through a 0.2 µm filter to obtain sterile protein, and the concentration was determined using a Bradford assay. 

### 2.4. Animals

The experimental animals were male Balb/C mice (aged eight to ten weeks) obtained from the Royan Research Institute, Karaj, Iran. The mice were maintained for one week before experiments started, with appropriate access provided to food and water, and kept in a 12/12 h light/dark cycle. The animal investigations were undertaken according to the standards of the European Communities Council (86/609/EEC).

### 2.5. Experimental Groups and Immunization

The experimental mice were separated into three groups, consisting of 6 mice each. The first group was vaccinated with GPEHT protein (40 µg) created in the alum adjuvant, while the 2nd group was vaccinated with GPEHT protein (20 µg) developed in the alum adjuvant. In the control groups, PBS+Alum was created in the alum adjuvant and injected into the third group. All injections contained an overall quantity of 100 µL and were performed subcutaneously. Experimental groups were vaccinated three times on days 0, 14, and 28. Two weeks after the last immunization, the immunologic parameters were analyzed. In the preparation of hybrid proteins (40 and 20 µg) with 30 µL aluminum, hydroxide was combined prior to immunization and carefully rotated for 2 h to assist binding. After centrifugation, the concentration of unabsorbed protein in the supernatant was determined using a BCA assay kit. The results indicated a binding effectiveness of 96% between the alum and hybrid proteins. Subsequently, the prepared mixture of the hybrid protein and alum was utilized to immunize the mice.

### 2.6. Spleen Cell Suspension and Cytokine Analysis

Two weeks after the last shots, the spleens of the mice were gathered and suspended in PBS with 2% FBS. The suspension was centrifuged at 300 g at 4 °C for 5 min. The cell suspension was incubated with 5 mL red blood cell lysis buffer for 5 min. Then, 5 mL of RPMI-1640 (Gibco, Dreieich, Germany), consisting of 10% FBS, four mM L-glutamine, 100 µg/ mL streptomycin, and 100 IU/mL penicillin, was included in the last suspension and passed through a sieve tube. The suspension was washed three times with cold PBS and centrifuged at 300 g/5 min at 4 °C. The spleen suspension was prepared at a density of 3 × 105 and seeded in 24-well plates under sterilized conditions, then boosted with and without the GPEHT antigen at a concentration of 10 µg/ mL to collect the supernatants after 72 h of incubation. Eventually, supernatants were collected 72 h after incubation and kept at −70 °C until the cytokine assay. Two weeks after the last shot, the levels of IFN-γ, IL-4, and IL-17 cytokines were examined. Afterward, the collected supernatants were utilized to evaluate the levels of IFN-γ, IL-4, and IL-17 cytokines using an ELISA cytokine discovery kit (R&D Systems, Minneapolis, MN, USA). Each cytokine was evaluated as pg/mL based on the plotted standard deviation.

### 2.7. Specific Total IgG, IgG1, and IgG2a Isotypes

Fourteen days after the last vaccination dose (day 42), peripheral blood was gathered from the mice to assess the stimulated antibody responses via an enzyme-linked immunosorbent assay (ELISA) in the immunized mouse groups. The maximized indirect ELISA was utilized to assess total IgG, IgG1, and IgG2a antibodies in vaccinated and non-vaccinated sera according to the following protocol. In short, 96-well ELISA plates (Greiner, Frickenhausen, Germany) were covered with purified GPEHT protein (10 μg/ mL) in a coating buffer (0.1 mM NaHCO_3_, pH = 7.4). Subsequently, the coated plates were blocked with bovine serum albumin (BSA; Sigma) in PBS. After that, sera were diluted in a blocking buffer (1:100 to 1/51,200), placed in the wells, and incubated for 2 h at 37 °C. In addition, after washing the plates five times, 100 µL per well of HRP-conjugated anti-mouse IgG was used as secondary antibodies, diluted 1:10,000 (Sigma). After washing, the plates were incubated in the dark for 30 min with 100 L per well of tetramethylbenzidine (TMB) substrate. The response was ended with 2NH_2_SO_4_, and absorbance was assessed at 450 nm with an ELISA reader. The same approach used to measure total IgG was also used to measure the isotype antibodies (IgG1 and IgG2a). Nevertheless, after adding mouse serum, secondary antibodies specific for every single isotype were utilized to identify the isotype antibodies.

### 2.8. Statistical Analysis

In the present research, data are shown as the mean ± S.D. Graphs were produced using GraphPad (GraphPad Program Inc., San Diego, CA, USA). The information from the study was evaluated via Student’s *t*-test and one-way analysis of variance (ANOVA) with Tukey’s multiple-comparison examination for numerous mouse groups. A value of *p* < 0.05 was considered statistically significant.

## 3. Results

### 3.1. Codon Optimization and Cloning

The hybrid GPEHT protein was designed based on the HTLV-1 Gag, Pol, Env, Hbz, and Tax proteins using bioinformatics tools (Figure 1). The evaluations showed that the GPEHT protein is an antigen, not an allergen, and its tertiary structure is close to the structure of proteins in nature [28]. The general characteristics of the GPEHT-designed protein are presented in Table 1. The codon optimization of the DNA sequence for *E. coli* codon usage was carried out by Biomatik Company with the OPTIMIZER server. The gene was constructed into the NcoI and XhoI sites of the expression vector pET28a with a poly-histidine tag (6x-His tag) to generate a protein with a 6x-His tag at the C-terminus.

### 3.2. Expression, Purification, and Characterization of the GPEHT Recombinant Protein

The transformation of the plasmid construct was done with E. coli BL21 (DE3) using the heat shock method. The GPEHT fusion protein was successfully expressed and purified using a nickel chromatography column. To confirm the existence of the GPEHT cloned gene, plasmids were isolated from two positive clones using a QIAGEN plasmid extraction and purification kit. Subsequently, using T7 polymerase-specific primers (T7 promoter primer, TAATACGACTCACTATAGGG and T7 terminator primer, GCTAGTTATTGCTCAGCG), it was confirmed via PCR that the size of the gene was about 2000 bases (Figure 2). The evaluation of the GPEHT purified protein was performed using SDS-PAGE and Western blot analysis with the His-tag antibody and showed single bands of approximately 70 kDa for this protein, which closely matched the calculated values (Figure 2). The LAL test indicated less than 0.5 EU/mL levels of LPS in the purified and dialyzed proteins. Moreover, the Appendix A showed the densitometry intensity ratio of each band for the SDS-PAGE and Western blot.

### 3.3. Analysis of the Humoral Immune Responses

The ability of the vaccine candidates to elicit a humoral response was assessed in mice by measuring the total IgG antibodies against the GPEHT protein using the indirect ELISA method (Figure 3). Total IgG was measured in different serum dilutions (1:100–1:6400) 2 weeks after the second and third immunizations. The results indicated that a 1:200 dilution offered the best differentiation between the IgG levels of the mouse groups. The antibody responses to the GPEHT protein in concentrations of 20 and 40 µg were significantly enhanced compared to those of the control mice that received PBS+Alum (*p* < 0.0001). In addition, mice vaccinated with 40 μg of GPEHT protein showed a slight increase compared to mice that received 20 μg of this protein. The total antibody concentration did not show a significant increase 28 days after the injection of the vaccine candidate, and it showed a significant increase only 42 days after the injection.

### 3.4. Measurement of Antibody Isotype Responses

The isotype antibodies IgG1 and IgG2a were evaluated in the serum 2 weeks after the third vaccination to assess the quality of the immune reactions. All mouse groups generated significant IgG1 and IgG2a reactions versus the GPEHT protein compared to the control mice (*p* < 0.05). The adjuvant did not significantly improve the IgG1 or IgG2a reactions to the GPEHT protein (*p* > 0.05) (Figure 4). In general, the IgG2a levels were higher than those of IgG1 in the groups administered 20 and 40 µg of the GPEHT antigen. The formula containing 40 µg GPEHT protein with the alum adjuvant boosted the levels of IgG2a and IgG1 isotypes compared to the 20 µg GPEHT protein developed with the alum adjuvant.

### 3.5. IL-17, IL-4, and IFN-γ Cytokines of Experimental Mice

In the present investigation, cytokine levels were measured to identify the T-cell-dependent immunity caused by the vaccination candidates. The mice vaccinated with the alum–GPEHT protein (40 µg) and 20 µg GPEHT protein vaccine candidates significantly enhanced the levels of IFN-γ, IL-4, and IL-17. In addition, the 40 µg GPEHT protein concentration was observed to yield further stimulation of the immune system and a stronger response (Figure 5A–C).

The evaluation of the Th1-dependent cytokine (IFN-γ) responses showed that GPEHT protein can successfully activate cellular immunity. Additionally, a dramatic increase in the Th17-dependent cytokine response was observed in the group of mice that received GPEHT protein (40 µg) in comparison to other groups (*p* < 0.001), based on exposure of the spleen cell suspension to this protein. Moreover, a comparison of the ratio of IFN-γ to IL-4 was carried out. As shown in Figure 5D, the ratio of IFN-γ/IL-4 was significantly (*p* < 0.001) higher in mice immunized with the GPEHT protein in comparison with the control group. The result indicates that the ratio of IFN-γ to IL-4 is much higher.

## 4. Discussion

Developing a prophylactic and therapeutic vaccine against viral, bacterial, and parasitic infections can save millions of lives [29]. Although different groups are seeking an effective vaccine against HTLV-1, to date, there remains no efficient vaccine for HTLV-1 infection [30]. Peptide-based vaccines have vital benefits such as safety, lower expenses, and high efficiency [31]. However, creating a reliable peptide vaccine has difficulties, such as the reduced innate immunogenicity of each epitope [32]. Additionally, generating a single-epitope peptide vaccine against HTLV-1 might result in inadequate immune reactions. Thus, based on our previous report and unlike the previous candidate vaccine, which relied on a portion of the HTLV-1 genome, in this work we proposed a recombinant peptide-based vaccine originating from different parts of the HTLV-1 genome to induce a high immune response in mice [28,33]. Moreover, several subunit vaccines failed to provide preventive and consistent immune actions. For that reason, subunit vaccines need an appropriate adjuvant and delivery system for improved efficiency and selective delivery to immune cells [34]. In this experiment, we used alum as an adjuvant due to its capabilities, such as increasing antigen uptake to induce robust humoral immunity.

In addition, an efficient vaccine against viral infections should generate cellular or humoral immunity. In HTLV-1 infections, various authors have recommended that both cellular and humoral immunity are needed to develop a reliable protective immune reaction [33]. A particular antibody can prevent the virus from entering target cells [35]. Moreover, cellular immunity can clear infections [36]. On the basis of the previous report, a GPEHT multiepitope chimeric protein from different parts of HTLV-1 was used to generate a high level of immune reactions identified by B and T cells without autoimmune reactions [28]. The GPEHT chimera protein was placed in a plasmid vector to be expressed in the bacterial host. Subsequently, affinity chromatography was applied to purify the expressed chimera protein to vaccinate the mice.

It was previously shown that the primary reservoir of HTLV-1 is CD4+ T cells [37]. On the basis of their features, CD4+ T cells were divided into Th1 and Th2 subsets. Activation of Th1 cells primarily creates the IFN-γ cytokine, which is essential for the clearance of intracellular microorganisms [38]. IFN-γ is a pleiotropic cytokine produced by stimulated T cells and natural killer (NK) cells [39]. This cytokine plays a primary role in the innate and adaptive aspects of host antiviral and antitumor immune protection, acting in various ways on host and tumor cells to favor tumor regression by generating cellular antitumor immunity, inflammation, cell cycle arrest, and apoptosis, as well as hindering angiogenesis and stimulating antigen presentation through the upregulation of major histocompatibility complex (MHC) I and II [40]. Additionally, previous reports showed that IFN-γ results in anti-tumor effector features that might regulate the initiation, development, or spread of Tax-transgenic tumors in mice [41].

In comparison, Th2 cells can generate the IL-4 cytokine, which is responsible for extracellular pathogens, humoral immunity, eosinophils activation, and antibody manufacturing [38,42]. Nevertheless, different investigations also showed that the overproduction of IL-4 can have a negative effect on antiviral activity against HTLV-1 infection [43].

This research used a chimera vaccine formulated with an alum adjuvant to enhance the production of Th1 and Th2 cytokines and related specific antibodies. In the groups vaccinated with 20 and 40 μg of the GPEHT protein, the production of IFN-γ and IL-4 was significantly higher than in the control group. However, the essential part of the Th1 action shifted the Th1/Th2 balance toward Th1 in both groups. Additionally, previous reports on mice showed that more Th1/Th2 negatively correlated with the proviral load of HTLV-1 in animal models. Thus, a weak to modest Th2 immune reaction is needed to generate a suitable immune action against HTLV-1 infection and stop Th1 hypersensitivity [33]. In addition, our previous investigation and other reports showed that a high IFN-γ response is essential to protect the body from viral infection. For instance, a recent study by Kabiri et a1. (2018) showed that a high IFN-γ response is essential for anti-inflammatory responses and could be necessary for the clearance of HTLV-1 infection [38]. Moreover, the higher ratio of IFN-γ to IL-4 indicates the tendency of the immune response to be stronger in cellular immunity than in humoral immunity. Since the main response of the immune system to viral infection is cellular immunity, it can be concluded that the immune response generated against the GPEHT protein is competent.

Additionally, vaccination with a chimeric vaccine was found to induce a high IL-17 cytokine response. IL-17 is a proinflammatory cytokine associated with the induction of inflammation [44]. In HTLV-1-contaminated people, this cytokine can act synergistically with Th1 in favor of inflammation [45,46,47]. Notably, a high amount of IL-17 can stimulate the production of other inflammatory cytokines such as TNF-α and IL-6, leading to tissue damage [44,48]. Moreover, the previous investigation showed that in patients with HAM/TSP disease, the secretion of IL-17 is less than that in uninfected controls [45]. The results showed that in the vaccinated group, the levels of IL-17 increased significantly compared to those in the control group. Therefore, aiming at the bifunctional activity of IL-17, more investigations are needed to analyze the protective activity of IL-17 in mice challenged with HTLV-1 infection [49].

Additionally, based upon antibody assays, subcutaneous vaccination with the chimera vaccine promoted a high degree of IgG1 and IgG2a isotypes. In mice, Th1 and Th2 immune reactions were found to be connected to the induction of IgG2a and IgG1 antibodies, respectively [41]. This increase in the isotype-specific immune response can play an essential role in eliminating the virus [41]. Moreover, previous reports and our investigation showed that high antibody responses could induce a high protective effect against HTLV-1 infection. For instance, a recent study by Shafifar et al. (2022) showed that high antibody production can negatively correlate with the proviral load of HTLV-1 [33]. Additionally, a study by Kabiri et al. (2018) showed that high IgG2a and IgG1 antibodies can induce TH1 and TH2 activation and offer strong protective effects against HTLV-1 infection [41].

This investigation has some limitations, including the lack of a challenge-based experiment and no analysis of anti-HTLV-1-specific antibodies to see if the immune responses generated this particular humoral immunity.

## 5. Conclusions

This examination aimed to evaluate the performance of a unique GPEHT protein vaccine consisting of immunogenic epitopes of HTLV-1 linked with a flexible linker in the presence of adjuvants to boost humoral and cellular immunity. The outcomes suggested that administration of the GPEHT chimera vaccine with the alum adjuvant evoked a high immune response and shifted Th1/Th2 towards durable Th1 immunity. Moreover, the developed vaccine minimized the degree of immunosuppressive cytokines such as IL-17 in addition to manufacturing proinflammatory cytokines and boosting systemic immune reactions.

## Figures and Tables

**Figure 1 vaccines-11-00115-f001:**
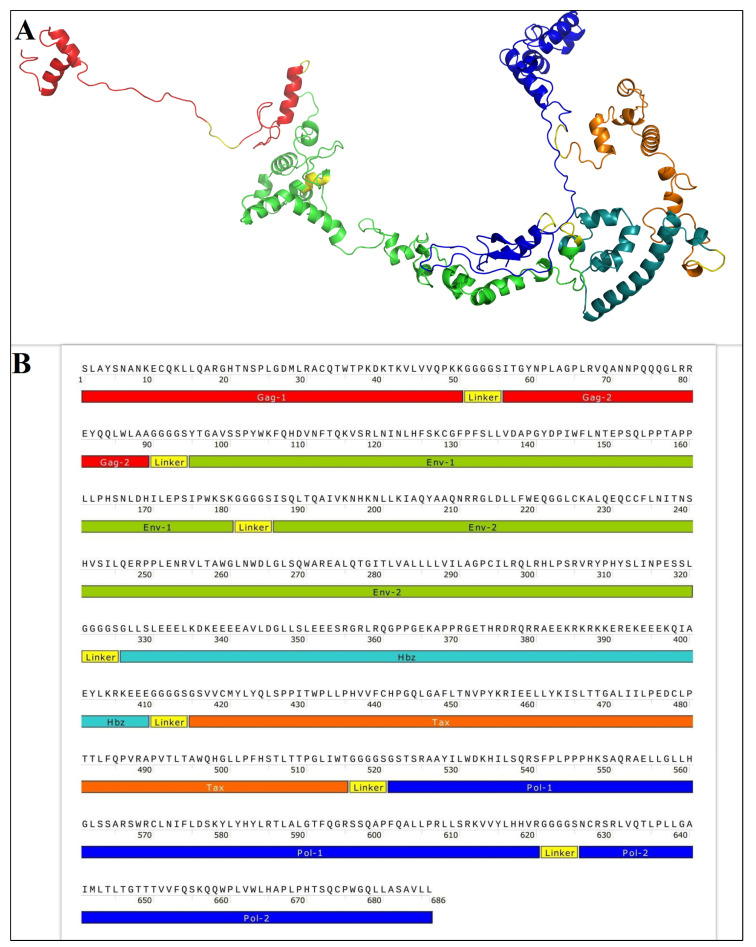
(**A**) The tertiary structure of the GPEHT protein. (**B**) A schematic diagram of the final construct of the vaccine candidate.

**Figure 2 vaccines-11-00115-f002:**
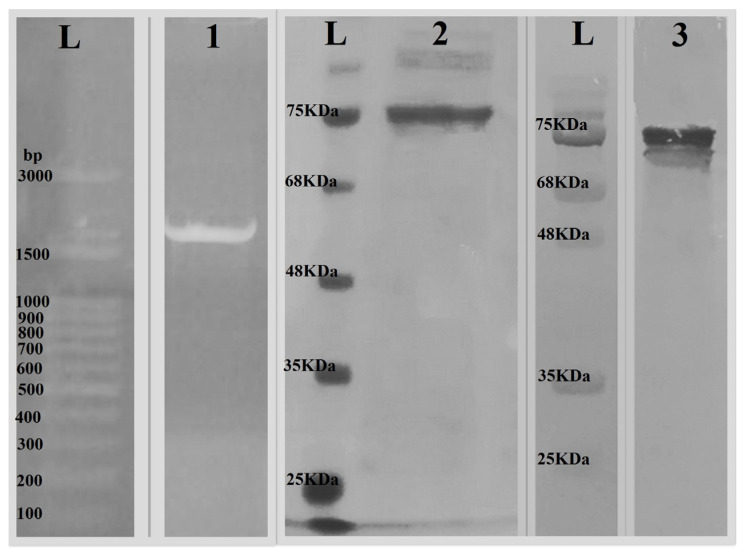
(1) Confirmation of the presence of the GPEHT gene in the transformed colonies (~2000 bp); (2) SDS PAGE, purification of GPEHT protein (~70 kDa); and (3) Western blot of GPEHT protein (~70 kDa). L: Ladder.

**Figure 3 vaccines-11-00115-f003:**
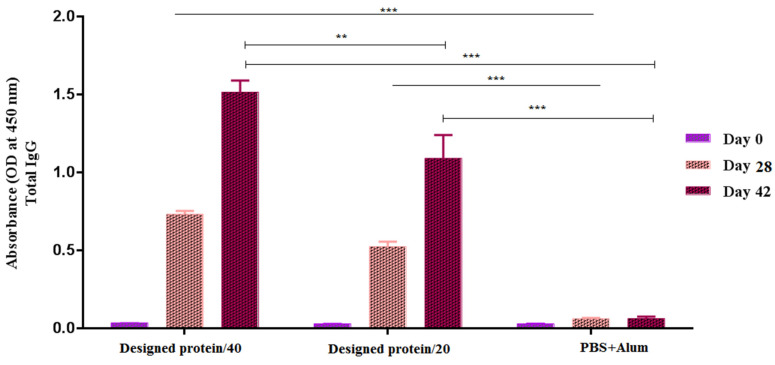
Specific total IgG in experimental groups with indirect ELISA against GPEHT protein. A significant increase was observed in the mice immunized with the GPEHT protein (20 and 40 µg candidates) compared to the PBS+Alum control group. *** for *p* < 0.0001. ** for *p* < 0.001.

**Figure 4 vaccines-11-00115-f004:**
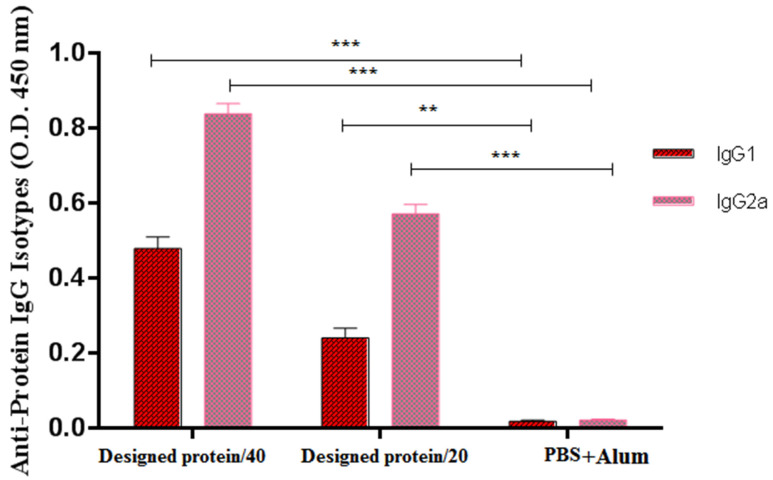
The GPEHT protein with 20 and 40 µg candidates induced high titers of IgG1 and IgG2a. The 40-µg protein candidate induced higher levels of specific IgG1 and IgG2a compared to the 20-µg protein candidate. *** for *p* < 0.0001; ** for *p* < 0.001.

**Figure 5 vaccines-11-00115-f005:**
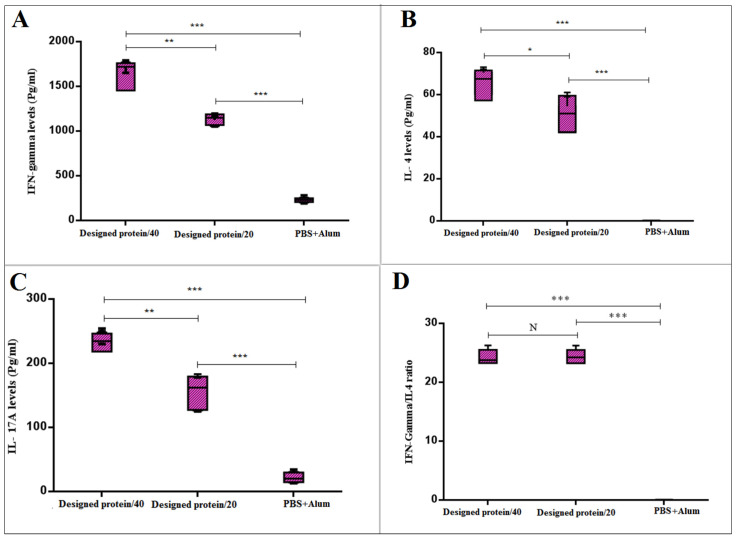
Analysis of the cytokine response in the immunized mice with 20 and 40 µg of GPEHT protein candidates and PBS+Alum. Mice were sacrificed 14 days after the third injection. Splenocytes from each mouse group were prepared and promoted with the GPEHT protein for 72 h, and the supernatants were evaluated for the secretion of (**A**) IFN-γ, (**B**) IL-4, and (**C**) IL-17 levels. (**D**) IFN-γ/IL-4 ratio. Outcomes are the mean stimulation index ± S.D. of six mice per group from three independent experiments. *** for *p* < 0.0001, ** for *p* < 0.001, and * for *p* < 0.01. N for No Significant Difference.

**Table 1 vaccines-11-00115-t001:** The physicochemical parameters, antigenicity, and allergenicity of GPEHT protein.

Characteristics	GPEHT
Number of amino acids	686
Molecular weight (dalton)	76.1598
Total number of negatively charged residues (Asp + Glu)	51
Total number of positively charged residues (Arg + Lys)	75
Theoretical isoelectric point (pI)	9.51
Estimated half-life	1.9 h in mammalian reticulocytes, >20 h in yeast, and >10 h in *E. coli*
Aliphatic index	94.27
Grand average hydropathy	−0.283
Antigenicity	0.4885
Allergenicity	Non-allergenic

## Data Availability

The authors acknowledge that the data presented in this study must be deposited and made publicly available in an acceptable repository prior to publication. The frontiers cannot accept a manuscript that does not adhere to our open data policies.

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
