# Peer review of "Recombinant GPEHT Fusion Protein Derived from HTLV-1 Proteins with Alum Adjuvant Induces a High Immune Response in Mice"

_vaccines, 2023, doi:10.3390/vaccines11010115_

Round 1

Reviewer 1 Report

The work is important although, as addressed by the authors themselves at the end of the manuscript, there was no challenge or evaluation of the induction of anti-HTLV antibodies. In this sense, I suggest modifying the title of the work, since the induction of response against HTLV was not evaluated, but against the vaccine protein. In addition, it would be interesting to evaluate the induction of IL10 and Treg cells, since previous works cited by the authors suggest that this type of response would not be protective against HTLV-1 infection.

Attention should be given to the first paragraph of the introduction, as all sentences begin with HTLV-1, making the grammar monotonous.

Lines 66 and 67, from reported.. is meaningless...

Page 71 recombinant vaccine instead of recombinant vaccination would be more appropriate

How many experiments were performed? Only 1?

Modify the term PBS by Alum or PBS+Alum in the charts

Regarding total IgG, a difference was observed between antigen concentrations only 42 days after immunization, this should be explained in the text. Although there is a tendency towards greater production of IgG1 and IgG2 in the group immunized with 40ug, this difference is not statistically significant.

As the authors themselves report the importance of the Th1/Th2 ratio in the outcome of the infection, I suggest that they carry out an assessment of the IFNg/IL4 ratio as a way of evaluating this parameter between different vaccine concentrations. The same could be done for IgG2a and IgG1

Lines 321 and 323 note the word "additionally" repeated at the beginning of two sentences in a row

Author Response

The work is important although, as addressed by the authors themselves at the end of the manuscript, there was no challenge or evaluation of the induction of anti-HTLV antibodies. In this sense, I suggest modifying the title of the work, since the induction of response against HTLV was not evaluated, but against the vaccine protein. In addition, it would be interesting to evaluate the induction of IL10 and Treg cells, since previous works cited by the authors suggest that this type of response would not be protective against HTLV-1 infection.

  • Dear reviewer, Thank you for these important comments regarding the title, We corrected it in the revised version. Also regarding IL-10 and Treg cells, We agree, it could be very interesting to assay IL-10 and Treg, however in this project first we try to prove the ability of recombinant vaccines to induce different aspects of immune responses including TH1, TH2, and TH 17. we will in the next project surely add this important cytokine for the assay of vaccine efficacy.

Attention should be given to the first paragraph of the introduction, as all sentences begin with HTLV-1, making the grammar monotonous.

  • We should thank you for this comment. The first paragraph and also all introductions were revised to be more clear and easy to follow.

Lines 66 and 67, from reported. is meaningless...

  • The sentence corrected.

Page 71 recombinant vaccine instead of recombinant vaccination would be more appropriate

  • The sentence corrected.

How many experiments were performed? Only 1?

  • To check the immune response, we examined the experimental group of recombinant protein in two different doses of 20 and 40 micrograms. This study aims to investigate the effect of this protein in provoking antibodies and IL-4, IL-17, and IFN-Gamma cytokines, which can be considered a candidate for future studies. In some studies, a group is selected as the first study to evaluate the immune system response to protein candidates [1, 2].

Modify the term PBS by Alum or PBS+Alum in the charts

  • PBS modified by PBS+Alum

Regarding total IgG, a difference was observed between antigen concentrations only 42 days after immunization, this should be explained in the text. Although there is a tendency towards greater production of IgG1 and IgG2 in the group immunized with 40ug, this difference is not statistically significant.

  • Requested changes were made in the manuscript.

As the authors themselves report the importance of the Th1/Th2 ratio in the outcome of the infection, I suggest that they carry out an assessment of the IFNγ/IL4 ratio as a way of evaluating this parameter between different vaccine concentrations. The same could be done for IgG2a and IgG1

  • Requested changes were made in the manuscript.

Lines 321 and 323 note the word "additionally" repeated at the beginning of two sentences in a row

  • Requested changes were made in the manuscript.

Refrences

  1. Ahmadi, K., et al., Epitope-based immunoinformatics study of a novel Hla-MntC-SACOL0723 fusion protein from Staphylococcus aureus: Induction of multi-pattern immune responses.Molecular immunology, 2019. 114: p. 88-99.
  2. Farshidi, N., et al., Preparation and pre-clinical evaluation of flagellin-adjuvanted NOM vaccine candidate formulated with Spike protein against SARS-CoV-2 in mouse model.Microbial Pathogenesis, 2022. 171: p. 105736.

Reviewer 2 Report

The title can be improved such as "Recombinant GPEHT fusion protein with alum adjuvant induces a high immune response in mice". The term "against HTLV-1" is not valid because the study does not include an evaluation using virus. The sera obtained from vaccinated mice could be used to perform an antiviral assay. 

Line 90: The gene or protein sequence accession numbers should be indicated.

Line 100: The transformation technique is " heat shock" 

Line 170: The title could be better as "statistical analysis"

Fig.s 3, 4 and 5: The control group is not "PBS". The group received PBS+alum.

The names in Latin should be written italic throughout the text.

Author Response

The title can be improved such as "Recombinant GPEHT fusion protein with alum adjuvant induces a high immune response in mice". The term "against HTLV-1" is not valid because the study does not include an evaluation using virus. The sera obtained from vaccinated mice could be used to perform an antiviral assay. 

  • Thank you for this important comments , The title revised with your valuable advice .

Line 90: The gene or protein sequence accession numbers should be indicated.

  • Requested changes were made in the manuscript.

Line 100: The transformation technique is " heat shock" 

  • Requested changes were made in the manuscript.

Line 170: The title could be better as "statistical analysis"

  • Requested changes were made in the manuscript.

Fig.s 3, 4 and 5: The control group is not "PBS". The group received PBS+alum.

  • Requested changes were made in the manuscript.

The names in Latin should be written italic throughout the text.

  • Requested changes were made in the manuscript.

Reviewer 3 Report

Comments to the Authors   Manuscript ID: vaccines-2019786 Title: Vaccination of mice with GPEHT recombination protein with alum adjuvant induces a high immune response against HTLV-1 Authors: Hamid Reza Jahantigh *, Angela Stufano, Farhad Koohpeyma, Vijihe Sadat Nikbin, Zahra Shahosseini, Piero Lovreglio

This manuscript by Jahantigh et al reports on a fusion protein, GPEHT protein, which derived from HTLV-1 structural and non-structural proteins, as a candidate for vaccine to prevent HTLV-1 infection. The authors observed 70-kDa protein expression by Western blot, and purified production was administrated to mice with adjuvant for three times. By the administration of the recombinant protein, total IgG was elevated and both of IgG1 and IgG2a isotypes production were induced. Splenocytes extracted from vaccinated mice released high-level IFN-gamma, IL-4 and IL-17 via coculture with GPEHT protein for 72 hours. The associated diseases of HTLV-1, ATL and HAM, have not been curable yet, and the preventive measure for HTLV-1 infection is one of the most effective tools for reduction of burden of ATL and HAM. However, the authors do not indicate the characteristics of the expressed proteins and any neutralizing activities against HTLV-1 infection in this manuscript.

1. Abstract.

The last sentence of the abstract is unclear. The authors mentioned the GPEHT protein could be a prospective vaccine candidate for defense against HTLV-1. This data was not presented in the article result section where it should be. Fig.2 only showed that the product from transformant was His-tagged 70-kDa protein. And no data on neutralization ability of induced antibody (IgG) fraction by administration of the obtained protein.

2. Materials and methods.

The authors showed PBS administration as a control in experiments, however, the administration of obtained sample from mock (vector-only) transformant is necessary for the evidence of the GPEHT protein function. Additionally, the authors should clarify that the obtained 70-kDa protein has the amino acid sequence derived from HTLV-1 proteins, using specific antibody for each component of fusion protein or fragment detection by LC-MS-MS.

The authors should use superscript and subscript adequately, for example, L.144 3 x 105, L.159-160 NaHCO3, L.166 2NH2SO4, and so on.

L.170 What is ‘Analytical analysis’?

3. Results.

As described above, the author should indicate the evidence of purified product from transformant has accurate amino acid sequence of HTLV-1-derived fusion protein according to their DNA sequence, in addition to the effect of neutralization for HTLV-1 infection by the product.

The authors did not show the necessity of supplementary file anywhere.

L.217-218 The authors did not show any results on this description.

L.247-249 The authors mentioned ‘no significant induction of IL-17’ in this sentence, however, in the Abstract, the authors described significant increase of IL-17 as same as IFN-gamma and IL-4. In Figure 5C, the significance was shown with asterisks.

4. Figures

Figure 2-5 The authors did not describe the meaning of asterisks accurately.

In the legend of figure 5, mice were sacrificed 12 days after the third injection, namely, day 40, while, in Materials and Methods, the spleens of the mice were gathered two weeks after the last shots, namely, day 42. Which is correct?

5. References

Many references of this manuscript are not adequate for the evidence the authors referred.

For example, in the paper referred as ref.1 demonstrated the phenomenon of T-cell exhaustion in both HTLV-1 AC and ATL patients. The authors of ref. 1 indicated that the continuous exposure of Tax-CTL to HTLV-1-infected cells attenuate their function, potentially leading to the development of ATL. This paper is not adequate for the reference which shows HTLV-1 as the cause of ATL onset.

I recommend the reconsideration of references below: ref. 1,2, 3, 4, 5, 10, 23, 24, 37, 41, 43 and 45.

Author Response

Comments to the Authors   Manuscript ID: vaccines-2019786 Title: Vaccination of mice with GPEHT recombination protein with alum adjuvant induces a high immune response against HTLV-1 Authors: Hamid Reza Jahantigh *, Angela Stufano, Farhad Koohpeyma, Vijihe Sadat Nikbin, Zahra Shahosseini, Piero Lovreglio

 This manuscript by Jahantigh et al reports on a fusion protein, GPEHT protein, which derived from HTLV-1 structural and non-structural proteins, as a candidate for vaccine to prevent HTLV-1 infection. The authors observed 70-kDa protein expression by Western blot, and purified production was administrated to mice with adjuvant for three times. By the administration of the recombinant protein, total IgG was elevated and both of IgG1 and IgG2a isotypes production were induced. Splenocytes extracted from vaccinated mice released high-level IFN-gamma, IL-4 and IL-17 via coculture with GPEHT protein for 72 hours. The associated diseases of HTLV-1, ATL and HAM, have not been curable yet, and the preventive measure for HTLV-1 infection is one of the most effective tools for reduction of burden of ATL and HAM. However, the authors do not indicate the characteristics of the expressed proteins and any neutralizing activities against HTLV-1 infection in this manuscript.

  1. Abstract.

The last sentence of the abstract is unclear. The authors mentioned the GPEHT protein could be a prospective vaccine candidate for defense against HTLV-1. This data was not presented in the article result section where it should be. Fig.2 only showed that the product from transformant was His-tagged 70-kDa protein. And no data on neutralization ability of induced antibody (IgG) fraction by administration of the obtained protein.

  • Thank you for the careful opinion of the respected reviewer. The last sentence was corrected.

  1. Materials and methods.

The authors showed PBS administration as a control in experiments, however, the administration of obtained sample from mock (vector-only) transformant is necessary for the evidence of the GPEHT protein function. Additionally, the authors should clarify that the obtained 70-kDa protein has the amino acid sequence derived from HTLV-1 proteins, using specific antibody for each component of fusion protein or fragment detection by LC-MS-MS.

  • Thank you for the point mentioned by the honorable reviewer. According to the experimental groups investigated in the previous studies [1-4], as a control, we used the basic buffer in which the vaccine candidate is formulated and includes salt with alum adjuvant. If the honorable reviewer means the cell transformed with the empty vector: in the process of purifying the empty vector, only the elution buffer is left, which contains imidazole and urea, and finally, after dialysis with PBS, Imidazole and urea are replaced by PBS, and The buffer that remains is the same PBS that we inject into the control group.
  • This study is the first step in our research to evaluate the efficacy of this vaccine candidate. We have planned this phase of our study based on previous studies. Our candidate protein was synthesized by Biomatik company and then, sequencing was done to confirm the sequence of our protein by the company, after transferring the vector containing the protein to the expression cell, the expression was confirmed by PAGE and western blot with an HRP-conjugated His-specific antibody. This method has been used in many studies to confirm the His-tagged protein. We can examine the animal phase in different studies [1-3, 5],

The authors should use superscript and subscript adequately, for example, L.144 3 x 105, L.159-160 NaHCO3, L.166 2NH2SO4, and so on.

  • It was corrected in the text.

L.170 What is ‘Analytical analyses?

  • It was corrected in the text.
  1. Results.

As described above, the author should indicate the evidence of purified product from transformant has accurate amino acid sequence of HTLV-1-derived fusion protein according to their DNA sequence, in addition to the effect of neutralization for HTLV-1 infection by the product.

  • Sequencing of our designed construct after synthesis was done by the Biomatik company. In this study, the confirmation of the expressed protein has been done using HRP-conjugated His-specific antibody against the designed protein as with many past studies [1-3].

The authors did not show the necessity of supplementary file anywhere.

  • We add in line 201-202 in revised version .

L.217-218 The authors did not show any results on this description.

  • The desired sentence was corrected.

L.247-249 The authors mentioned ‘no significant induction of IL-17’ in this sentence, however, in the Abstract, the authors described significant increase of IL-17 as same as IFN-gamma and IL-4. In Figure 5C, the significance was shown with asterisks.

  • Our writing was wrong. The sentence was corrected in the manuscript.
  1. Figures

Figure 2-5 The authors did not describe the meaning of asterisks accurately.

  • The necessary change was applied.

In the legend of figure 5, mice were sacrificed 12 days after the third injection, namely, day 40, while, in Materials and Methods, the spleens of the mice were gathered two weeks after the last shots, namely, day 42. Which is correct?

  • It was corrected in the manuscript.

  1. References

Many references of this manuscript are not adequate for the evidence the authors referred.

For example, in the paper referred as ref.1 demonstrated the phenomenon of T-cell exhaustion in both HTLV-1 AC and ATL patients. The authors of ref. 1 indicated that the continuous exposure of Tax-CTL to HTLV-1-infected cells attenuate their function, potentially leading to the development of ATL. This paper is not adequate for the reference which shows HTLV-1 as the cause of ATL onset. I recommend the reconsideration of references below: ref. 1,2, 3, 4, 5, 10, 23, 24, 37, 41, 43 and 45.

  • We should thank you so much for this important comment, we critically changes the references based your comment and highlighted in main text .

References

  1. Oloomi, M., et al., Protective multi-epitope candidate vaccine for urinary tract infection.Biotechnology Reports, 2020. 28: p. e00564.
  2. Fathi, M., et al., Formulation of a recombinant HIV‐1 polytope candidate vaccine with naloxone/alum mixture: Induction of multi‐cytokine responses with a higher regulatory mechanism.Apmis, 2021. 129(8): p. 480-488.
  3. Ahmadi, K., et al., Epitope-based immunoinformatics study of a novel Hla-MntC-SACOL0723 fusion protein from Staphylococcus aureus: Induction of multi-pattern immune responses.Molecular immunology, 2019. 114: p. 88-99.
  4. Farshidi, N., et al., Preparation and pre-clinical evaluation of flagellin-adjuvanted NOM vaccine candidate formulated with Spike protein against SARS-CoV-2 in mouse model.Microbial Pathogenesis, 2022. 171: p. 105736.
  5. Hasanzadeh, S., et al., In silico analysis and in vivo assessment of a novel epitope-based vaccine candidate against uropathogenic Escherichia coli.Scientific reports, 2020. 10(1): p. 1-16.

Reviewer 4 Report

The study describes the development of the multiepitope vaccine GPEHT combined from five HTLV-1 proteins Gag, Pol, Env, HBZ, and Tax, and linked via GGGGS peptide. The polypeptide was produced in E. coli., purified, and subcutaneously injected into mice in combination with Alum adjuvant. The immunogenicity of the vaccine-formulated preparation was estimated by measuring the level of IFNgamma, IL-4, and IL-17 in the culture of splenocytes from immunized mice after restimulation with GPEHT, and by measuring the levels of anti-GPEHT IgG (IgG1 and IgG2a) in mouse peripheral blood samples. Although a B- and T-cell immune response in mice to the total protein GPEHT was shown, this study had no attempts to demonstrate any anti-HTLV-1 immune responses, either protective or non-protective. Therefore, the title of the paper overstates the significance of the presented results, and there are conclusions that are not supported by the data. As such, the paper cannot be recommended for publication. Below, I outlined the major drawbacks of the paper:

1.     Different multiepitope vaccines had been developed against HIV in the past but they have demonstrated low protective properties. Therefore, for each vaccine, it is important to show antiviral effects. As both HIV and HTLV Envs are heavily N-glycosylated proteins, immunization with vaccine protein produced in bacteria often results in low binding and neutralizing capacity of vaccinal antibodies. In order to demonstrate the generation of anti-HTLV-1 Abs, at least it is necessary to test GPEHT immune sera for binding with HTLV-1 Env expressed in a mammalian cell line. More strong and convincing results can be obtained using any of the HTLV-1 neutralizing tests (for instance, described in J Virol. 2001 Sep; 75(18): 8461–8468,  PLoS Pathog. 2010 Feb; 6(2): e1000788)

2.     Instead of bulk T-cell response to GPEHT polypeptide, it is important to show cytokine production (IFN) after splenocyte stimulation with immunodominant peptides from each Gag, Pol, Hbz, and Tax genes included in the polypeptide. This would prove that T cells are responding to viral peptides and not to the sequences that overlap with linker amino acids, which are not protective. Although it is difficult to translate the mouse response to the human one, at least it gives more information about which T-cell epitopes are more immunogenic in GPEHT.

3.     The text contains a lot of awkward phrases like “hindering HBZ decreases the proviral load and leads to the absence of manufacturing HTLV-1-specific antibodies”, “..via the warm shock technique”, “the spleens of the mice were gathered and suspended in PBS”. The manuscript should be corrected by an English professional biologist.

4.     Some statements and conclusions are not supported by the results: the Title “..high immune response to HTLV-1”; lines 61-62 “After HTLV-1 infects a cell, the disease's development occurs through cell-to-cell transmission” is a wrong statement since cell-to-cell transmission is the only way for this infection, cell-free serum, and other biological liquors do not transmit HTLV-1; lines 243-244 “GPEHT protein can successfully activate cellular immunity to eliminate the virus”.

5.     Lines 58-61. There is no reference to scientists who discovered GLUT1 receptor for HTLV-1 (Cell. 2003 Nov 14;115(4):449-59)

Author Response

The study describes the development of the multiepitope vaccine GPEHT combined from five HTLV-1 proteins Gag, Pol, Env, HBZ, and Tax, and linked via GGGGS peptide. The polypeptide was produced in E. coli., purified, and subcutaneously injected into mice in combination with Alum adjuvant. The immunogenicity of the vaccine-formulated preparation was estimated by measuring the level of IFNgamma, IL-4, and IL-17 in the culture of splenocytes from immunized mice after restimulation with GPEHT, and by measuring the levels of anti-GPEHT IgG (IgG1 and IgG2a) in mouse peripheral blood samples. Although a B- and T-cell immune response in mice to the total protein GPEHT was shown, this study had no attempts to demonstrate any anti-HTLV-1 immune responses, either protective or non-protective. Therefore, the title of the paper overstates the significance of the presented results, and there are conclusions that are not supported by the data. As such, the paper cannot be recommended for publication. Below, I outlined the major drawbacks of the paper:

  1. Different multiepitope vaccines had been developed against HIV in the past but they have demonstrated low protective properties. Therefore, for each vaccine, it is important to show antiviral effects. As both HIV and HTLV Envs are heavily N-glycosylated proteins, immunization with vaccine protein produced in bacteria often results in low binding and neutralizing capacity of vaccinal antibodies. In order to demonstrate the generation of anti-HTLV-1 Abs, at least it is necessary to test GPEHT immune sera for binding with HTLV-1 Env expressed in a mammalian cell line. More strong and convincing results can be obtained using any of the HTLV-1 neutralizing tests (for instance, described in J Virol. 2001 Sep; 75(18): 8461–8468,  PLoS Pathog. 2010 Feb; 6(2): e1000788)
  • Thank you for these important comments, This study is the first phase of our investigation to evaluate the antigenicity and ability of this protein candidate to stimulate the immune system in mice. As Tan et. al. have produced the influenza fusion protein as a vaccine candidate in the Ecoli bacterial expression system and examined its function. Production of the fusion protein in E. coli BL21 (DE3) promises a rapid, less tedious, and cost-effective protein expression system [1]. We have used an env protein domain in the protein design. And this protein is a protein with a new design including Gag, Pol, Env, HBZ, and Tax proteins. In some (1-8) studies, E.coli cell has also been used for expression of the HIV and HTLV-1 protein vaccine candidate. According to the very worthy suggestion of the honorable referee, we will continue to study the expression of this protein in mammalian cells in future studies and the next phase.

  1. Instead of bulk T-cell response to GPEHT polypeptide, it is important to show cytokine production (IFN) after splenocyte stimulation with immunodominant peptides from each Gag, Pol, Hbz, and Tax genes included in the polypeptide. This would prove that T cells are responding to viral peptides and not to the sequences that overlap with linker amino acids, which are not protective. Although it is difficult to translate the mouse response to the human one, at least it gives more information about which T-cell epitopes are more immunogenic in GPEHT.
  • Our aim of this study was to select epitope-rich domains of different proteins of HTLV-1. Because the vaccine candidate in the form of protein can maintain the conformation of the protein domain to some extent and also cover more epitopes. On the other hand, in recent studies, protein candidates have received more attention than peptide candidates [1-3]. Individual epitopes can be used to stimulate spleen cells, and this is a very good suggestion that can be used in further studies. According to most previous studies, we used protein antigens that contain epitope-rich domains to stimulate spleen cells [3-7].

  1. The text contains a lot of awkward phrases like “hindering HBZ decreases the proviral load and leads to the absence of manufacturing HTLV-1-specific antibodies”, “..via the warm shock technique”, “the spleens of the mice were gathered and suspended in PBS”. The manuscript should be corrected by an English professional biologist.

  • we should thank you for these important comments, before submitting we submit the article for English review by MDPI editing services, However, we again revised it with one professional biologist and immunologist to be more clear and easy to follow. 

  1. Some statements and conclusions are not supported by the results: the Title “..high immune response to HTLV-1”; lines 61-62 “After HTLV-1 infects a cell, the disease's development occurs through cell-to-cell transmission” is a wrong statement since cell-to-cell transmission is the only way for this infection, cell-free serum, and other biological liquors do not transmit HTLV-1; lines 243-244 “GPEHT protein can successfully activate cellular immunity to eliminate the virus”.

  • Thank you for this critical comment, L 61-62 corrected in main text .
  • Regarding the sentence cell-to-cell transmission …., the main text corrected .
  • We should tThank you for this critical comment, L 243-244 corrected in main text .
  1. Lines 58-61. There is no reference to scientists who discovered GLUT1 receptor for HTLV-1 (Cell. 2003 Nov 14;115(4):449-59)
  • Thank you for your comments, We in the revised version correct the references and also updates some other references, and highlighted them in main text.

References

  1. Ahmadi, K., et al., Epitope-based immunoinformatics study of a novel Hla-MntC-SACOL0723 fusion protein from Staphylococcus aureus: Induction of multi-pattern immune responses.Molecular immunology, 2019. 114: p. 88-99.
  2. Farshidi, N., et al., Preparation and pre-clinical evaluation of flagellin-adjuvanted NOM vaccine candidate formulated with Spike protein against SARS-CoV-2 in mouse model.Microbial Pathogenesis, 2022. 171: p. 105736.
  3. Oloomi, M., et al., Protective multi-epitope candidate vaccine for urinary tract infection.Biotechnology Reports, 2020. 28: p. e00564.
  4. Fathi, M., et al., Formulation of a recombinant HIV‐1 polytope candidate vaccine with naloxone/alum mixture: Induction of multi‐cytokine responses with a higher regulatory mechanism.Apmis, 2021. 129(8): p. 480-488.
  5. Habibi, M., et al., Immunization with recombinant protein Ag43:: UpaH with alum and 1, 25 (OH) 2D3 adjuvants significantly protects Balb/C mice against urinary tract infection caused by uropathogenic Escherichia coli.International Immunopharmacology, 2021. 96: p. 107638.
  6. Hasanzadeh, S., et al., In silico analysis and in vivo assessment of a novel epitope-based vaccine candidate against uropathogenic Escherichia coli.Scientific reports, 2020. 10(1): p. 1-16.
  7. Tan, M.P., et al., Expression of Influenza M2e-NP Recombinant Fusion Protein in Escherichia coli BL21 (DE3) and Its Binding to Antibodies.Vaccines, 2022. 10(12): p. 2066.
  8. Malonis, R.J., J.R. Lai, and O. Vergnolle, Peptide-based vaccines: current progress and future challenges.Chemical reviews, 2019. 120(6): p. 3210-3229.
  9. Arashkia, A., et al., Severe acute respiratory syndrome‐coronavirus‐2 spike (S) protein based vaccine candidates: State of the art and future prospects.Reviews in medical virology, 2021. 31(3): p. e2183.
  10. Escalona, E., D. Sáez, and A. Oñate, Immunogenicity of a multi-epitope dna vaccine encoding epitopes from Cu–Zn superoxide dismutase and open reading Frames of Brucella abortus in mice.Frontiers in immunology, 2017. 8: p. 125.

Round 2

Reviewer 1 Report

The authors modified the document and added suggestions from both reviewers. I have no further considerations and consider that the manuscript is ready for publication.

Author Response

Dear reviewer

We are so grateful for helping us to improve our article.

warm regards 

Reviewer 4 Report

The authors addressed my minor revision comments. None of the requested experiments have been set up. Since the title of the paper was changed, and any conclusions about antiviral protection of created vaccine were eliminated, there is no need for anti-HTLV-1 immune response measurement. However, it should be noted, that this significantly reduces the merit of the presented paper that can be taken into consideration for this paper.

Following the revision, the English was improved, although a roughness is still present in the text, just for example in Fig.5 legend (“Examination of the cytokine actions in the immunized mice…” could be rephrased). The anti-HTLV immune response was deleted from the title, and that is correct, as explained above. However, it remains totally unclear what GPEHT is and against what? Hence, the title should be modified by indicating that the formulation GPEHT was derived from HTLV-1 proteins.

Author Response

Following the revision, the English was improved, although a roughness is still present in the text, just for example in Fig.5 legend (“Examination of the cytokine actions in the immunized mice…” could be rephrased). The anti-HTLV immune response was deleted from the title, and that is correct, as explained above. However, it remains totally unclear what GPEHT is and against what? Hence, the title should be modified by indicating that the formulation GPEHT was derived from HTLV-1 proteins.

Dear reviewer

We are so grateful for helping us to improve our article. The title was modified regarding the suggestion.

Also mentioned sentences rewrite to be clearer.